# NON-VACUOUS GENERALIZATION BOUNDS AT THE IMAGENET SCALE: A PAC-BAYESIAN COMPRESSION APPROACH

**Wenda Zhou**
Columbia University
New York, NY
wz2335@columbia.edu

**Victor Veitch**
Columbia University
New York, NY
victorveitch@gmail.com

**Morgane Austern**
Columbia University
New York, NY
ma3293@columbia.edu

**Ryan P. Adams**
Princeton University
Princeton, NJ
rpa@princeton.edu

**Peter Orbanz**
Columbia University
New York, NY
porbanz@stat.columbia.edu

## ABSTRACT

Modern neural networks are highly overparameterized, with capacity to substantially overfit to training data. Nevertheless, these networks often generalize well in practice. It has also been observed that trained networks can often be "compressed" to much smaller representations. The purpose of this paper is to connect these two empirical observations. Our main technical result is a generalization bound for compressed networks based on the compressed size that, combined with off-the-shelf compression algorithms, leads to state-of-the-art generalization guarantees. In particular, we provide the first non-vacuous generalization guarantees for realistic architectures applied to the ImageNet classification problem. Additionally, we show that compressibility of models that tend to overfit is limited. Empirical results show that an increase in overfitting increases the number of bits required to describe a trained network.

## 1 INTRODUCTION

A pivotal question in machine learning is why deep networks perform well despite overparameterization. These models often have many more parameters than the number of examples they are trained on, which enables them to drastically overfit to training data (Zhang et al., 2017a). In common practice, however, such networks perform well on previously unseen data.

Explaining the generalization performance of neural networks is an active area of current research. Attempts have been made at adapting classical measures such as VC-dimension (Harvey et al., 2017) or margin/norm bounds (Neyshabur et al., 2018; Bartlett et al., 2017), but such approaches have yielded bounds that are vacuous by orders of magnitudes. Other authors have explored modifications of the training procedure to obtain networks with provable generalization guarantees (Dziugaite & Roy, 2017; 2018). Such procedures often differ substantially from standard procedures used by practitioners, and empirical evidence suggests that they fail to improve performance in practice (Wilson et al., 2017).

We begin with an empirical observation: it is often possible to "compress" trained neural networks by finding essentially equivalent models that can be described in a much smaller number of bits; see Cheng et al. (2018) for a survey. Inspired by classical results relating small model size and generalization performance (often known as *Occam's razor*), we establish a new generalization bound based on the effective compressed size of a trained neural network. Combining this bound with off-the-shelf compression schemes yields the first non-vacuous generalization bounds in practical problems. The main contribution of the present paper is the demonstration that, unlike many other measures, this measure is effective in the deep-learning regime.

Generalization bound arguments typically identify some notion of complexity of a learning problem, and bound generalization error in terms of that complexity. Conceptually, the notion of complexity we identify is:

$$\text{complexity} = \text{compressed size} - \text{remaining structure}. \tag{1}$$

The first term on the right-hand side represents the link between generalization and explicit compression. The second term corrects for superfluous structure that remains in the compressed representation. For instance, the predictions of trained neural networks are often robust to perturbations of the network weights. Thus, a representation of a neural network by its weights carries some irrelevant information. We show that accounting for this robustness can substantially reduce effective complexity.

Our results allow us to derive explicit generalization guarantees using off-the-shelf neural network compression schemes. In particular:

- The generalization bound can be evaluated by compressing a trained network, measuring the effective compressed size, and substituting this value into the bound.
- Using off-the-shelf neural network compression schemes with this recipe yields bounds that are state-of-the-art, including the first non-vacuous bounds for modern convolutional neural nets.

The above result takes a compression algorithm and outputs a generalization bound on nets compressed by that algorithm. We provide a complementary result by showing that if a model tends to overfit then there is an absolute limit on how much it can be compressed. We consider a classifier as a (measurable) function of a random training set, so the classifier is viewed as a random variable. We show that the entropy of this random variable is lower bounded by a function of the expected degree of overfitting. Additionally, we use the randomization tests of Zhang et al. (2017a) to show empirically that increased overfitting implies worse compressibility, for a fixed compression scheme.

The relationship between small model size and generalization is hardly new: the idea is a variant of Occam's razor, and has been used explicitly in classical generalization theory (Rissanen, 1986; Blumer et al., 1987; MacKay, 1992; Hinton & van Camp, 1993; Rasmussen & Ghahramani, 2001). However, the use of highly overparameterized models in deep learning seems to directly contradict the Occam principle. Indeed, the study of generalization and the study of compression in deep learning has been largely disjoint; the later has been primarily motivated by computational and storage limitations, such as those arising from applications on mobile devices (Cheng et al., 2018). Our results show that Occam type arguments remain powerful in the deep learning regime. The link between compression and generalization is also used in work by Arora et al. (2018), who study compressibility arising from a form of noise stability. Our results are substantially different, and closer in spirit to the work of Dziugaite & Roy (2017); see Section 3 for a detailed discussion.

Zhang et al. (2017a) study the problem of generalization in deep learning empirically. They observe that standard deep net architectures—which generalize well on real-world data—are able to achieve perfect training accuracy on randomly labelled data. Of course, in this case the test error is no better than random guessing. Accordingly, any approach to controlling generalization error of deep nets must selectively and preferentially bound the generalization error of models that are actually plausible outputs of the training procedure applied to real-world data. Following Langford & Caruana (2002); Dziugaite & Roy (2017); Neyshabur et al. (2018), we make use of the PAC-Bayesian framework (McAllester, 1999; Catoni, 2007; McAllester, 2013). This framework allows us to encode prior beliefs about which learned models are plausible as a (prior) distribution $\pi$ over possible parameter settings. The main challenge in developing a bound in the PAC-Bayes framework bound is to articulate a distribution $\pi$ that encodes the relative plausibilities of possible outputs of the training procedure. The key insight is that, implicitly, any compression scheme is a statement about model plausibilities: good compression is achieved by assigning short codes to the most probable models, and so the probable models are those with short codes.

## 2 GENERALIZATION AND THE PAC-BAYESIAN PRINCIPLE

In this section, we recall some background and notation from statistical learning theory. Our aim is to learn a classifier using data examples. Each example $(x, y)$ consists of some features $x \in \mathcal{X}$ and a label $y \in \mathcal{Y}$. It is assumed that the data are drawn identically and independently from some data generating distribution, $(X_i, Y_i) \overset{iid}{\sim} \mathcal{D}$. The goal of learning is to choose a hypothesis $h : \mathcal{X} \to \mathcal{Y}$

that predicts the label from the features. The quality of the prediction is measured by specifying some loss function $L$; the value $L(h(x), y)$ is a measure of the failure of hypothesis $h$ to explain example $(x, y)$. The overall quality of a hypothesis $h$ is measured by the risk under the data generating distribution:

$$L(h) = \mathbb{E}_{(X,Y) \sim \mathcal{D}}[L(h(X), Y)].$$

Generally, the data generating distribution is unknown. Instead, we assume access to training data $S_n = \{(x_1, y_1), \ldots, (x_n, y_n)\}$, a sample of $n$ points drawn i.i.d. from the data generating distribution. The true risk is estimated by the empirical risk:

$$\hat{L}(h) = \frac{1}{n} \sum_{(x,y) \in S} L(h(x), y).$$

The task of the learner is to use the training data to choose a hypothesis $\hat{h}$ from among some pre-specified set of possible hypothesis $\mathcal{H}$, the hypothesis class. The standard approach to learning is to choose a hypothesis $\hat{h}$ that (approximately) minimizes the empirical risk. This induces a dependency between the choice of hypothesis and the estimate of the hypothesis' quality. Because of this, it can happen that $\hat{h}$ overfits to the training data: $\hat{L}(\hat{h}) \ll L(\hat{h})$. The generalization error $L(\hat{h}) - \hat{L}(\hat{h})$ measures the degree of overfitting. In this paper, we consider an image classification problem, where $x_i$ is an image and $y_i$ the associated label for that image. The selected hypothesis is a deep neural network. We mostly consider the 0 -1 loss, that is, $L(h(x), y) = 0$ if the prediction is correct and $L(h(x), y) = 1$ otherwise.

We use the PAC-Bayesian framework to establish bounds on generalization error. In general, a PAC-Bayesian bound attempts to control the generalization error of a stochastic classifier by measuring the discrepancy between a pre-specified random classifier (often called *prior*), and the classifier of interest. Conceptually, PAC-Bayes bounds have the form:

$$\text{generalization error of } \rho \leq O\left(\sqrt{\text{KL}(\rho, \pi)/n}\right), \tag{2}$$

where $n$ is the number of training examples, $\pi$ denotes the prior, and $\rho$ denotes the classifier of interest (often called *posterior*).

More formally, we write $L(\rho) = \mathbb{E}_{h \sim \rho}[L(h)]$ for the risk of the random estimator. The fundamental bound in PAC-Bayesian theory is (Catoni, 2007, Thm. 1.2.6):

**Theorem 2.1** (PAC-Bayes). *Let $L$ be a $\{0, 1\}$-valued loss function, let $\pi$ be some probability measure on the hypothesis class, and let $\alpha > 1, \epsilon > 0$. Then, with probability at least $1 - \epsilon$ over the distribution of the sample:*

$$L(\rho) \leq \inf_{\lambda > 1} \Phi_{\lambda/n}^{-1} \left\{ \hat{L}(\rho) + \frac{\alpha}{\lambda} \left[ \text{KL}(\rho, \pi) - \log \epsilon + 2 \log\left(\frac{\log(\alpha^2 \lambda)}{\log \alpha}\right) \right] \right\}, \tag{3}$$

*where we define $\Phi_\gamma^{-1}$ as:*

$$\Phi_\gamma^{-1}(x) = \frac{1 - e^{-\gamma x}}{1 - e^{-\gamma}}. \tag{4}$$

*Remark* 2.2. The above formulation of the PAC-Bayesian theorem is somewhat more opaque than other formulations (e.g., McAllester, 2003; 2013; Neyshabur et al., 2018). This form is significantly tighter when $\text{KL}/n$ is large. See Bégin et al. (2014); Laviolette (2017) for a unified treatment of PAC-Bayesian bounds.

The quality of a PAC-Bayes bound depends on the discrepancy between the PAC-Bayes prior $\pi$—encoding the learned models we think are plausible—and $\rho$, which is the actual output of the learning procedure. The main challenge is finding good choices for the PAC-Bayes prior $\pi$, for which the value of $\text{KL}(\rho, \pi)$ is both small and computable.

## 3 RELATIONSHIP TO PREVIOUS WORK

**Generalization.** The question of which properties of real-world networks explain good generalization behavior has attracted considerable attention (Langford, 2002; Langford & Caruana, 2002; Hinton & van Camp, 1993; Hochreiter & Schmidhuber, 1997; Baldassi et al., 2015; 2016; Chaudhari et al., 2017; Keskar et al., 2017; Dziugaite & Roy, 2017; Schmidt-Hieber, 2017; Neyshabur et al.,

2017; 2018; Arora et al., 2018); see Arora (2017) for a review of recent advances. Such results typically identify a property of real-world networks, formalize it as a mathematical definition, and then use this definition to prove a generalization bound. Generally, the bounds are very loose relative to the true generalization error, which can be estimated by evaluating performance on held-out data. Their purpose is not to quantify the actual generalization error, but rather to give qualitative evidence that the property underpinning the generalization bound is indeed relevant to generalization performance. The present paper can be seen in this tradition: we propose compressibility as a key signature of performant real-world deep nets, and we give qualitative evidence for this thesis in the form of a generalization bound.

The idea that compressibility leads to generalization has a long history in machine learning. Minimum description length (MDL) is an early formalization of the idea (Rissanen, 1986). Hinton & van Camp (1993) applied MDL to very small networks, already recognizing the importance of weight quantization and stochasticity. More recently, Arora et al. (2018) consider the connection between compression and generalization in large-scale deep learning. The main idea is to compute a measure of noise-stability of the network, and show that it implies the existence of a simpler network with nearly the same performance. A variant of a known compression bound (see (McAllester, 2013) for a PAC-Bayesian formulation) is then applied to bound the generalization error of this simpler network in terms of its code length. In contrast, the present paper develops a tool to leverage existing neural network compression algorithms to obtain strong generalization bounds. The two papers are complementary: we establish non-vacuous bounds, and hence establish a quantitative connection between generalization and compression. An important contribution of Arora et al. (2018) is obtaining a quantity measuring the compressibility of a neural network; in contrast, we apply a compression algorithm and witness its performance. We note that their compression scheme is very different from the sparsity-inducing compression schemes (Cheng et al., 2018) we use in our experiments. Which properties of deep networks allow them to be sparsely compressed remains an open question.

To strengthen a naïve Occam bound, we use the idea that deep networks are insensitive to mild perturbations of their weights, and that this insensitivity leads to good generalization behavior. This concept has also been widely studied (e.g., Langford, 2002; Langford & Caruana, 2002; Hinton & van Camp, 1993; Hochreiter & Schmidhuber, 1997; Baldassi et al., 2015; 2016; Chaudhari et al., 2017; Keskar et al., 2017; Dziugaite & Roy, 2017; Neyshabur et al., 2018). As we do, some of these papers use a PAC-Bayes approach (Langford & Caruana, 2002; Dziugaite & Roy, 2017; Neyshabur et al., 2018). Neyshabur et al. (2018) arrive at a bound for non-random classifiers by computing the tolerance of a given deep net to noise, and bounding the difference between that net and a stochastic net to which they apply a PAC-Bayes bound. Like the present paper, Langford & Caruana (2002); Dziugaite & Roy (2017) work with a random classifier given by considering a normal distribution over the weights centered at the output of the training procedure. We borrow the observation of Dziugaite & Roy (2017) that the stochastic network is a convenient formalization of perturbation robustness.

The approaches to generalization most closely related to ours are, in summary:

| Reference | Structure | Non-Vacuous | |
| | | MNIST | ImageNet |
| --- | --- | --- | --- |
| Dziugaite & Roy (2017) | Perturbation Robustness | ✓ | ✗ |
| Neyshabur et al. (2018) | Perturbation Robustness | ✗ | ✗ |
| Arora et al. (2018) | Compressibility (from Perturbation Robustness) | ✗ | ✗ |
| Present paper | Compressibility and Perturbation Robustness | ✓ | ✓ |

These represent the best known generalization guarantees for deep neural networks. Our bound provides the first non-vacuous generalization guarantee for the ImageNet classification task, the *de facto* standard problem for which deep learning dominates. It is also largely agnostic to model architecture: we apply the same argument to both fully connected and convolutional networks. This is in contrast to some existing approaches that require extra analysis to extend bounds for fully connected networks to bounds for convolutional networks (Neyshabur et al., 2018; Konstantinos et al.; Arora et al., 2018).

**Compression.** The effectiveness of our work relies on the existence of good neural network compression algorithms. Neural network compression has been the subject of extensive interest in the last few years, motivated by engineering requirements such as computational or power constraints. We apply a relatively simple strategy in this paper in the line of Han et al. (2016), but we note that our bound is compatible with most forms of compression. See Cheng et al. (2018) for a survey of recent results in this field.

## 4 MAIN RESULT

We first describe a simple Occam's razor type bound that translates the quality of a compression into a generalization bound for the compressed model. The idea is to choose the PAC-Bayes prior $\pi$ such that greater probability mass is assigned to models with short code length. In fact, the bound stated in this section may be obtained as a simple weighted union bound, and a variation is reported in McAllester (2013). However, embedding this bound in the PAC-Bayesian framework allows us to combine this idea, reflecting the explicit compressible structure of trained networks, with other ideas reflecting different properties of trained networks.

We consider a non-random classifier by taking the PAC-Bayes posterior $\rho$ to be a point mass at $\hat{h}$, the output of the training (plus compression) procedure. Recall that computing the PAC-Bayes bound effectively reduces to computing $\mathrm{KL}(\rho, \pi)$.

**Theorem 4.1.** *Let $|h|_c$ denote the number of bits required to represent hypothesis $h$ using some pre-specified coding $c$. Let $\rho$ denote the point mass at the compressed model $\hat{h}$. Let $m$ denote any probability measure on the positive integers. There exists a prior $\pi_c$ such that:*

$$\mathrm{KL}(\rho, \pi_c) \leq |\hat{h}|_c \log 2 - \log(m(|\hat{h}|_c)). \tag{5}$$

This result relies only on the quality of the chosen coding and is agnostic to whether a lossy compression is applied to the model ahead of time. In practice, the code $c$ is chosen to reflect some explicit structure—e.g., sparsity—that is imposed by a lossy compression.

*Proof.* Let $\mathcal{H}_c \subseteq \mathcal{H}$ denote the set of estimators that correspond to decoded points, and note that $\hat{h} \in \mathcal{H}_c$ by construction. Consider the measure $\pi_c$ on $\mathcal{H}_c$:

$$\pi_c(h) = \frac{1}{Z} m(|h|_c) 2^{-|h|_c}, \text{ where } Z = \sum_{h \in \mathcal{H}_c} m(|h|_c) 2^{-|h|_c}. \tag{6}$$

As $c$ is injective on $\mathcal{H}_c$, we have that $Z \leq 1$. We may thus directly compute the KL-divergence from the definition to obtain the claimed result. $\square$

*Remark* 4.2. To apply the bound in practice, we must make a choice of $m$. A pragmatic solution is to simply consider a bound on the size of the model to be selected (e.g. in many cases it is reasonable to assume that the encoded model is smaller than $2^{64}$ bytes, which is $2^{72}$ bits), and then consider $m$ to be uniform over all possible lengths.

### 4.1 USING ROBUSTNESS TO WEIGHT PERTURBATIONS

The simple bound above applies to an estimator that is compressible in the sense that its encoded length with respect to some fixed code is short. However, such a strategy does not consider any structure on the hypothesis space $\mathcal{H}$. In practice, compression schemes will often fail to exploit some structure, and generalization bounds can be (substantially) improved by accounting for this fact. We empirically observe that trained neural networks are often tolerant to low levels of discretization of the trained weights, and also tolerant to some low level of added noise in the trained weights. Additionally, quantization is an essential step in numerous compression strategies (Han et al., 2016). We construct a PAC-Bayes bound that reflects this structure.

This analysis requires a compression scheme specified in more detail. We assume that the output of the compression procedure is a triplet $(S, C, Q)$, where $S = \{s_1, \dots, s_k\} \subseteq \{1, \dots, p\}$ denotes the location of the non-zero weights, $C = \{c_1, \dots, c_r\} \subseteq \mathbb{R}$ is a codebook,

and $Q = (q_1, \ldots, q_k), q_i \in \{1, \ldots, r\}$ denotes the quantized values. Most state-of-the-art compression schemes can be formalized in this manner (Han et al., 2016).

Given such a triplet, we define the corresponding weight $w(S, Q, C) \in \mathbb{R}^p$ as:

$$w_i(S, Q, C) = \begin{cases} c_{q_j} & \text{if } i = s_j, \\ 0 & \text{otherwise.} \end{cases} \tag{7}$$

Following Langford & Caruana (2002); Dziugaite & Roy (2017), we bound the generalization error of a stochastic estimator given by applying independent random normal noise to the non-zero weights of the network. Formally, we consider the (degenerate) multivariate normal centered at $w$: $\rho \sim \mathcal{N}(w, \sigma^2 J)$, with $J$ being a diagonal matrix such that $J_{ii} = 1$ if $i \in S$ and $J_{ii} = 0$ otherwise.

**Theorem 4.3.** *Let $(S, C, Q)$ be the output of a compression scheme, and let $\rho_{S,C,Q}$ be the stochastic estimator given by the weights decoded from the triplet and variance $\sigma^2$. Let $c$ denote some arbitrary (fixed) coding scheme and let $m$ denote an arbitrary distribution on the positive integers. Then, for any $\tau > 0$, there is some PAC-Bayes prior $\pi$ such that:*

$$\mathrm{KL}(\rho_{S,C,Q}, \pi) \leq (k\lceil \log r \rceil + |S|_c + |C|_c) \log 2 - \log m(k\lceil \log r \rceil + |S|_c + |C|_c)$$
$$+ \sum_{i=1}^{k} \mathrm{KL}\Big(\mathrm{Normal}(c_{q_i}, \sigma^2), \sum_{j=1}^{r} \mathrm{Normal}(c_j, \tau^2)\Big). \tag{8}$$

Note that we have written the KL-divergence of a distribution with a unnormalized measure (the last term), and in particular this term may (and often will) be negative. We defer the construction of the prior $\pi$ and the proof of Theorem 4.3 to the supplementary material.

*Remark* 4.4. We may obtain the first term $k\lceil \log r \rceil + |S|_c + |C|_c$ from the simple Occam's bound described in Theorem 4.1 by choosing the coding of the quantized values $Q$ as a simple array of integers of the correct bit length. The second term thus describes the adjustment (or number of bits we "gain back") from considering neighbouring estimators.

## 5 Generalization bounds in practice

In this section we present examples combining our theoretical arguments with state-of-the-art neural network compression schemes.[1] Recall that almost all other approaches to bounding generalization error of deep neural networks yield vacuous bounds for realistic problems. The one exception is Dziugaite & Roy (2017), which succeeds by retraining the network in order to optimize the generalization bound. We give two examples applying our generalization bounds to the models output by modern neural net compression schemes. In contrast to earlier results, this leads immediately to non-vacuous bounds on realistic problems. The strength of the Occam bound provides evidence that the connection between compressibility and generalization has substantive explanatory power.

We report $95\%$ confidence bounds based on the measured effective compressed size of the networks. The bounds are achieved by combining the PAC-Bayes bound Theorem 2.1 with Theorem 4.3, showing that $\mathrm{KL}(\rho, \pi)$ is bounded by the "effective compressed size". We note a small technical modification: we choose the prior variance $\tau^2$ layerwise by a grid search, this adds a negligible contribution to the effective size (see Appendix A.1 for the technical details of the bound).

**LeNet-5 on MNIST.** Our first experiment is performed on the MNIST dataset, a dataset of 60k grayscale images of handwritten digits. We fit the LeNet-5 (LeCun et al., 1998) network, one of the first convolutional networks. LeNet-5 has two convolutional layers and two fully connected layers, for a total of 431k parameters.

We apply a pruning and quantization strategy similar to that described in Han et al. (2016). We prune the network using Dynamic Network Surgery (Guo et al., 2016), pruning all but $1.5\%$ of the network weights. We then quantize the non-zero weights using a codebook with 4 bits. The location of the non-zero coordinates are stored in compressed sparse row format, with the index differences encoded using arithmetic compression.

---

[1]Code to reproduce the experiments is available in the supplementary material.

We consider the stochastic classifier given by adding Gaussian noise to each non-zero coordinate before each forward pass. We add Gaussian noise with standard deviation equal to $5\%$ of the difference between the largest and smallest weight in the filter. This results in a negligible drop in classification performance.We obtain a bound on the training error of $46\%$ (with $95\%$ confidence). The effective size of the compressed model is measured to be $6.23\,\text{KiB}$.

**ImageNet.** The ImageNet dataset (Russakovsky et al., 2015) is a dataset of about 1.2 million natural images, categorized into 1000 different classes. ImageNet is substantially more complex than the MNIST dataset, and classical architectures are correspondingly more complicated. For example, AlexNet (Krizhevsky et al., 2012) and VGG-16 (Simonoyan & Zisserman, 2014) contain 61 and 128 million parameters, respectively. Non-vacuous bounds for such models are still out of reach when applying our bound with current compression techniques. However, motivated by computational restrictions, there has been extensive interest in designing more parsimonious architectures that achieve comparable or better performance with significantly fewer parameters (Iandola et al., 2016; Howard et al., 2017; Zhang et al., 2017b). By combining neural net compression schemes with parsimonious models of this kind, we demonstrate a non-vacuous bounds on models with better performance than AlexNet.

Our simple Occam bound requires only minimal assumptions, and can be directly applied to existing compressed networks. For example, Iandola et al. (2016) introduce the SqueezeNet architecture, and explicitly study its compressibility. They obtain a model with better performance than AlexNet but that can be written in $0.47\,\text{MiB}$. A direct application of our naïve Occam bound yields non-vacuous bound on the test error of $98.6\%$ (with $95\%$ confidence). To apply our stronger bound—taking into account the noise robustness—we train and compress a network from scratch. We consider Mobilenet 0.5 (Howard et al., 2017), which in its uncompressed form has better performance and smaller size than SqueezeNet (Iandola et al., 2016).

Zhu & Gupta (2017) study pruning of MobileNet in the context of energy-efficient inference in resource-constrained environments. We use their pruning scheme with some small adjustments. In particular, we use Dynamic Network Surgery (Guo et al., 2016) as our pruning method but follow a similar schedule. We prune $67\%$ of the total parameters. The pruned model achieves a validation accuracy of $60\%$. We quantize the weights using a codebook strategy (Han et al., 2016). We consider the stochastic classifier given by adding Gaussian noise to the non-zero weights, with the variance set in each layer so as not to degrade our prediction performance. For simplicity, we ignore biases and batch normalization parameters in our bound, as they represent a negligible fraction of the parameters. We consider top-1 accuracy (whether the most probable guess is correct) and top-5 accuracy (whether any of the 5 most probable guesses is correct). The final "effective compressed size" is $350\,\text{KiB}$. The

Table 1: Summary of bounds obtained from compression

| Dataset | Orig. size | Comp. size | Robust. Adj. | Eff. Size | Error Bound | |
|---|---|---|---|---|---|---|
| | | | | | Top 1 | Top 5 |
| MNIST | $168.4\,\text{KiB}$ | $8.1\,\text{KiB}$ | $1.88\,\text{KiB}$ | $6.23\,\text{KiB}$ | $< 46\%$ | NA |
| ImageNet | $5.93\,\text{MiB}$ | $452\,\text{KiB}$ | $102\,\text{KiB}$ | $350\,\text{KiB}$ | $< 96.5\%$ | $< 89\%$ |

stochastic network has a top-1 accuracy of $65\%$ on the training data, and top-5 accuracy of $87\%$ on the training data. The small effective compressed size and high training data accuracy yield non-vacuous bounds for top-1 and top-5 test error. See Appendix B for the details of the experiment.

## 6 LIMITS ON COMPRESSIBILITY

We have shown that compression results directly imply generalization bounds, and that these may be applied effectively to obtain non-vacuous bounds on neural networks. In this section, we provide a complementary view: overfitting implies a limit on compressibility.

**Theory.** We first prove that the entropy of estimators that tend to overfit is bounded in terms of the expected degree of overfitting. That implies the estimators fail to compress on average. As previously, consider a sample $S_n = \{(x_1, y_1), \dots, (x_n, y_n)\}$ sampled i.i.d. from some distribution $\mathcal{D}$, and an

estimator (or selection procedure) $\hat{h}$, which we consider as a (random) function of the training data. The key observation is:

$$\mathbb{P}(L(\hat{h}(x), y) = 1 \mid (x, y) \in S_n) = \mathbb{E}(\hat{L}(\hat{h})),$$
$$\mathbb{P}(L(\hat{h}(x), y) = 1 \mid (x, y) \notin S_n) = \mathbb{E}(L(\hat{h})).$$

That is, the probability of misclassifying an example in the training data is smaller than the probability of misclassifying a fresh example, and the expected strength of this difference is determined by the expected degree of overfitting. By Bayes' rule, we thus see that the more $\hat{h}$ overfits, the better it is able to distinguish a sample from the training and testing set. Such an estimator $\hat{h}$ must thus "remember" a significant portion of the training data set, and its entropy is thus lower bounded by the entropy of its "memory".

**Theorem 6.1.** *Let $L$, $\hat{L}$, and $\hat{h}$ be as in the text immediately preceeding the theorem. For simplicity, assume that both the sample space $\mathcal{X} \times \mathcal{Y}$ and the hypothesis set $\mathcal{H}$ are discrete. Then,*

$$H(\hat{h}) \geq ng(\mathbb{E}[\hat{L}(\hat{h})], \mathbb{E}[L(\hat{h})]), \tag{9}$$

*where $g$ denotes some non-negative function (given explicitly in the proof).*

We defer the proof to the supplementary material.

**Experiments.** We now study this effect empirically. The basic tool is the randomization test of Zhang et al. (2017a): we consider a fixed architecture and a number of datasets produced by randomly relabeling the categories of some fraction of examples from a real-world dataset. If the model has sufficiently high capacity, it can be fit with approximately zero training loss on each dataset. In this case, the generalization error is given by the fraction of examples that have been randomly relabeled. We apply a standard neural net compression tool to each of the trained models, and we observe that the models with worse generalization require more bits to describe in practice.

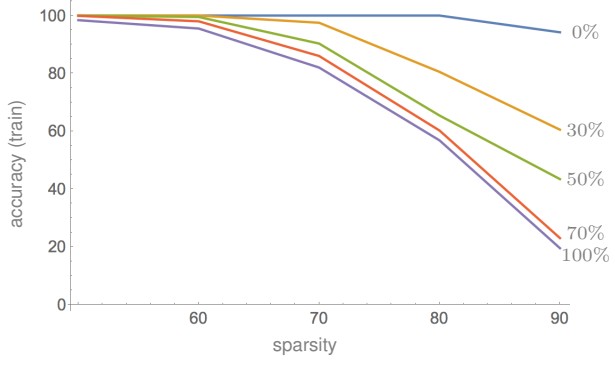

**Figure 1:** Training performance after pruning at varying levels of label randomization.

For simplicity, we consider the CIFAR-10 dataset, a collection of 40000 images categorized into 10 classes. We fit the ResNet (He et al., 2016) architecture with 56 layers with no pre-processing and no penalization on the CIFAR-10 dataset where the labels are subjected to varying levels of randomization. As noted in Zhang et al. (2017a), the network is able to achieve $100\,\%$ training accuracy no matter the level of randomization.

We then compress the networks fitted on each level of label randomization by pruning to a given target sparsity. Surprisingly, all networks are able to achieve $50\,\%$ sparsity with essentially no loss of training accuracy, even on completely random labels. However, we observe that as the compression level increases further, the scenarios with more randomization exhibit a faster decay in training accuracy, see Figure 1. This is consistent with the fact that network size controls generalization error.

## 7 DISCUSSION

It has been a long standing observation by practitioners that despite the large capacity of models used in deep learning practice, empirical results demonstrate good generalization performance. We show that with no modifications, a standard engineering pipeline of training and compressing a network leads to demonstrable and non-vacuous generalization guarantees. These are the first such results on networks and problems at a practical scale, and mirror the experience of practitioners that best

results are often achieved without heavy regularization or modifications to the optimizer (Wilson et al., 2017).

The connection between compression and generalization raises a number of important questions. Foremost, what are its limitations? The fact that our bounds are non-vacuous implies the link between compression and generalization is non-trivial. However, the bounds are far from tight. If significantly better compression rates were achievable, the resulting bounds would even be of practical value. For example, if a network trained on ImageNet to $90\%$ training and $70\%$ testing accuracy could be compressed to an effective size of $30\,\mathrm{KiB}$—about one order of magnitude smaller than our current compression—that would yield a sharp bound on the generalization error.

## 8 Acknowledgements

We acknowledge computing resources from Columbia University's Shared Research Computing Facility project, which is supported by NIH Research Facility Improvement Grant 1G20RR030893-01, and associated funds from the New York State Empire State Development, Division of Science Technology and Innovation (NYSTAR) Contract C090171, both awarded April 15, 2010

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

# A    PROOF OF THEOREM 4.3

In this section we describe the construction of the prior $\pi$ and prove the bound on the KL-divergence claimed in Theorem 4.3. Intuitively, we would like to express our prior as a mixture over all possible decoded points of the compression algorithm. More precisely, define the mixture component $\pi_{S,Q,C}$ associated with a triplet $(S, Q, C)$ as:

$$\pi_{S,Q,C} = \text{Normal}(w(S, Q, C), \tau^2). \tag{10}$$

We then define our prior $\pi$ as a weighted mixture over all triplets, weighted by the code length of the triplet:

$$\pi \propto \sum_{S,Q,C} m(|S|_c + |C|_c + k\lceil \log r \rceil) 2^{-|S|_c - |C|_c - k\lceil \log r \rceil} \pi_{S,Q,C}, \tag{11}$$

where the sum is taken over all $S$ and $C$ which are representable by our code, and all $Q = (q_1, \ldots, q_k) \in \{1, \ldots, r\}^k$. In practice, $S$ takes values in all possible subsets of $\{1, \ldots, p\}$, and $C$ takes values in $F^r$, where $F \subseteq \mathbb{R}$ is a chosen finite subset of representable real numbers (such as those that may be represented by IEEE-754 single precision numbers), and $r$ is a chosen quantization level. We now give the proof of Theorem 4.3.

*Proof.* We have that:

$$\pi = \frac{1}{Z} \sum_{S,Q,C} m(|S|_c + |C|_c + k\lceil \log r \rceil) 2^{-|S|_c - |C|_c - k\lceil \log r \rceil} \pi_{S,Q,C}, \tag{12}$$

where we must have $Z \leq 1$ by the same argument as in the proof of Theorem 4.1

Suppose that the output of our compression algorithm is a triplet $(\hat{S}, \hat{Q}, \hat{C})$. We recall that our posterior $\rho$ is given by a normal centered at $w(\hat{S}, \hat{Q}, \hat{C})$ with variance $\sigma^2$, and we may thus compute the KL-divergence:

$$
\begin{aligned}
\text{KL}(\rho, \pi) &\leq \text{KL}\left(\rho, \sum_{S,Q,C} m(|S|_c + |C|_c + k\lceil \log r \rceil) 2^{-|S|_c - |C|_c - k\lceil \log r \rceil} \pi_{S,Q,C}\right) \\
&\leq \text{KL}\left(\rho, \sum_Q m(|\hat{S}|_c + |\hat{C}|_c + \hat{k}\lceil \log \hat{r} \rceil) 2^{-|\hat{S}|_c - |\hat{C}|_c - \hat{k}\lceil \log \hat{r} \rceil} \pi_{\hat{S},Q,\hat{C}}\right) \\
&\leq \left(|\hat{S}|_c + |\hat{C}|_c + \hat{k}\lceil \log \hat{r} \rceil\right) \log 2 + \log m(|\hat{S}|_c + |\hat{C}|_c + \hat{k}\lceil \log \hat{r} \rceil) + \text{KL}\left(\rho, \sum_Q \pi_{\hat{S},Q,\hat{C}}\right).
\end{aligned}
\tag{13}
$$

We are now left with the mixture term, which is a mixture of $r^k$ many terms in dimension $k$, and thus computationally untractable. However, we note that we are in a special case where the mixture itself is independent across coordinates. Indeed, let $\phi_\tau$ denote the density of the univariate normal distribution with mean 0 and variance $\tau^2$, we note that we may write the mixture as:

$$
\begin{aligned}
\left(\sum_Q \pi_{\hat{S},Q,\hat{C}}\right)(x) &= \sum_{q^1,\ldots,q^k=1}^{r} \prod_{i=1}^{k} \phi_\tau(x_i - \hat{c}_{q^i}) \\
&= \prod_{i=1}^{k} \sum_{q^i=1}^{r} \phi_\tau(x_i - \hat{c}_{q^i}).
\end{aligned}
$$

Additionally, as our chosen stochastic estimator $\rho$ is independent over the coordinates, the KL-divergence decomposes over the coordinates, to obtain:

$$\text{KL}\left(\rho, \sum_Q \pi_{\hat{S},Q,\hat{C}}\right) = \sum_{i=1}^{k} \text{KL}\left(\rho_i, \sum_{q^i=1}^{r} \text{Normal}(\hat{c}_{q^i}, \tau^2)\right). \tag{14}$$

Plugging the above computation into (13) gives the desired result. $\qquad \square$

## A.1 DETAILS IN PRACTICAL USES OF THE BOUND

Although Theorem 4.3 contains the main mathematical contents of our bound, applying the bound in a fully correct fashion requires some amount of minutiae and book-keeping we detail in this section. In particular, we are required to select a number of parameters (such as the prior variances). We extend the bound to account for such unrestricted (and possibly data-dependent) parameter selection. Typically, such adjustments have a negligible effect on the computed bounds.

**Theorem A.1** (Union Bound for Discrete Parameters). *Let $\pi_\xi$, $\xi \in \Xi$, denote a family of priors parameterized by a discrete parameter $\xi$, which takes values in a finite set $\Xi$. There exists a prior $\pi$ such that for any posterior $\rho$ and any $\xi \in \Xi$:*

$$\mathrm{KL}(\rho, \pi) \leq \mathrm{KL}(\rho, \pi_\xi) + \log|\Xi|. \tag{15}$$

*Proof.* We define $\pi$ as a uniform mixture of the $\pi_\xi$:

$$\pi = \frac{1}{|\Xi|} \sum_{\xi \in \Xi} \pi_\xi. \tag{16}$$

We then have that:

$$\mathrm{KL}(\rho, \pi) = \mathbb{E}_{X \sim \rho} \log \frac{d\rho}{d\pi}, \tag{17}$$

but we can note that $\frac{d\rho}{d\pi} \leq |\Xi| \frac{d\rho}{d\pi_\xi}$, from which we deduce that:

$$\mathrm{KL}(\rho, \pi) \leq \mathrm{KL}(\rho, \pi_\xi) + \log|\Xi|. \tag{18}$$

$\square$

We make liberal use of this variant to control a number of discrete parameters which are chosen empirically (such as the quantization resolution at each layer). We also use this bound to control a number of continuous quantities (such as the prior variances) by discretizing these quantities as IEEE-754 single precision (32 bit) floating point numbers.

## B EXPERIMENT DETAILS

### B.1 LENET-5

We train the baseline model for LeNet-5 using stochastic gradient descent with momentum and no data augmentation. The batch size is set to 1024, and the learning rate is decayed using an inverse time decay starting at 0.01 and decaying every 125 steps. We apply a small $\ell_2$ penalty of 0.005. We train a total of 20000 steps.

We carry out the pruning using Dynamic Network Surgery (Guo et al., 2016). The threshold is selected per layer as the mean of the layer coefficients offset by a constant multiple of the standard deviation of the coefficients, where the multiple is piecewise constant starting at 0.0 and ending at 4.0. We choose the pruning probability as a piecewise constant starting at 1.0 and decaying to $10^{-3}$. We train for 30000 steps using the ADAM optimizer.

We quantize all the weights using a 4 bit codebook (Han et al., 2016) per layer initialized using $k$-means. A single cluster in each weight is given to be exactly zero and contains the pruned weights. The remaining clusters centers are learned using the ADAM optimizer over 1000 steps.

### B.2 MOBILENET

MobileNets are a class of networks that make use of depthwise separable convolutions. Each layer is composed of two convolutions, with one depthwise convolution and one pointwise convolution. We use the pre-trained MobileNet model provided by Google as our baseline model. We then prune the pointwise (and fully connected) layers only, using Dynamic Network Surgery. The threshold is set for each weight as a quantile of the absolute values of the coordinates, which is increased according to the schedule given in (Zhu & Gupta, 2017). As the lower layers are smaller and more sensitive, we

scale the target sparsity for each layer according to the size of the layer. The target sparsity is scaled linearly between $65\%$ and $75\%$ as a proportion of the number of elements in the layer compared to the largest layer (the final layer). We use stochastic gradient descent with momentum and decay the learning with an inverse time decay schedule, starting at $10^{-3}$ and decaying by $0.05$ every $2000$ steps. We use a minibatch size of $64$ and train for a total of $300000$ steps, but tune the pruning schedule so that the target sparsity is reached after $200000$ steps.

We quantize the weights by using a codebook for each layer with $6$ bits for all layers except the last fully connected layer which only has $5$ bits. The pointwise and fully connected codebooks have a reserved encoding for exact $0$, whereas the non-pruned depthwise codebooks are fully learned. We initialize the cluster assignment using $k$-means and train the cluster centers for $20000$ steps with stochastic gradient with momentum with a learning rate of $10^{-4}$. Note that we also modify the batch normalization moving average parameters in this step so that it adapts faster, choosing $.99$ as the momentum parameter for the moving averages.

To witness noise robustness, we only add noise to the pointwise and fully connected layer. We are able to add Gaussian noise with standard deviation equal to $2\%$ of the difference in magnitude between the largest and smallest coordinate in the layer for the fully connected layer. For pointwise layers we add noise equal to $1\%$ of the difference scaled linearly by the relative size of the layer compared to the fully connected layer. These quantities were chosen to minimally degrade the training performance while obtaining good improvements on the generalization bound: in our case, we observe that the top-1 training accuracy is reduced to $65\%$ with noise applied from $67\%$ without noise.

## C  PROOF THAT OVERFITTING IMPLIES HIGH CLASSIFIER ENTROPY

As previously, consider a sample $S = \{(x_1, y_1), \ldots, (x_n, y_n)\}$ sampled i.i.d. from some distribution $\mathcal{D}$, and an estimator (or selection procedure) $\hat{h}$. The statement that $\hat{h}$ overfits may then be captured in terms of the training and testing error of $\hat{h}$, namely that $\hat{L}(\hat{h}) \ll L(\hat{h})$. We note that this statement depends on the randomness of the sample through its impact on $\hat{h}$, and we will make the interpretation precise momentarily.

Such an estimator $\hat{h}$ that overfits may be transformed into a procedure which discriminates between samples from the training and testing set. Indeed, let $(x, y) \in \mathcal{X} \times \mathcal{Y}$ be drawn from an independent mixture of the uniform distribution on $S$ and the data-generating distribution $\mathcal{D}$, where by independent we mean that $I_{(x,y)\in S}$ is independent of $S$. Then, we have by Bayes rule that:

$$\mathbb{P}((x, y) \in S \mid L(\hat{h}(x), y) = 1) =$$
$$\frac{\mathbb{P}(L(\hat{h}(x), y) = 1 \mid (x, y) \in S)}{\mathbb{P}(L(\hat{h}(x), y) = 1 \mid (x, y) \in S) + \mathbb{P}(L(\hat{h}(x), y) = 1 \mid (x, y) \notin S)}, \quad (19)$$

where the probability is taken with respect to the distribution of $(x, y)$. By the definition of in-sample and out-of-sample loss, we have by independence that:

$$\mathbb{P}(L(\hat{h}(x), y) = 1 \mid (x, y) \in S) = \mathbb{E}(\hat{L}(\hat{h})),$$
$$\mathbb{P}(L(\hat{h}(x), y) = 1 \mid (x, y) \notin S) = \mathbb{E}(L(\hat{h})).$$

We may thus rewrite (19) (and its analogue conditional probability on the event $L(\hat{h}(x), y) = 0$) to obtain:

$$\mathbb{P}((x, y) \in S \mid L(\hat{h}(x), y) = 1) = \left(1 + \frac{\mathbb{E}(L(\hat{h}))}{\mathbb{E}(\hat{L}(\hat{h}))}\right)^{-1},$$
$$\mathbb{P}((x, y) \in S \mid L(\hat{h}(x), y) = 0) = \left(1 + \frac{1 - \mathbb{E}(L(\hat{h}))}{1 - \mathbb{E}(\hat{L}(\hat{h}))}\right)^{-1}.$$

We thus see that the more $\hat{h}$ overfits, the better it is able to distinguish a sample from the training and testing set. Such an estimator $\hat{h}$ must thus "remember" a significant portion of the training data set,

and its entropy is thus lower bounded by the entropy of its "memory". Quantitatively, we note that the quality of $\hat{h}$ as a discriminator between the training and testing set is captured by the quantities

$$p_n = \left(1 + \frac{\mathbb{E}(L(\hat{h}))}{\mathbb{E}(\hat{L}(\hat{h}))}\right)^{-1}, \quad q_n = \left(1 + \frac{1 - \mathbb{E}(L(\hat{h}))}{1 - \mathbb{E}(\hat{L}(\hat{h}))}\right)^{-1}, \quad l_n = \frac{1}{2}\mathbb{E}[\hat{L}(\hat{h}) + L(\hat{h})].$$

We may interpret $p_n$ as the average proportion of false positives and $q_n$ as the average proportion of true negatives when viewing $\hat{h}$ as a classifier. We prove that if those quantities are substantially different from a random classifier, then $\hat{h}$ must have high entropy. We formalize this statement and provide a proof below.

**Theorem C.1.** *Let $S = \{(x_1, y_1), \ldots, (x_n, y_n)\}$ be sampled i.i.d. from some distribution $\mathcal{D}$, and let $\hat{h}$ be a selection procedure, which is only a function of the unordered set $S$. Let us view $\hat{h}$ as a random quantity through the distribution induced by the sample $S$. For simplicity, we assume that both the sample space $\mathcal{X} \times \mathcal{Y}$ and the hypothesis set $\mathcal{H}$ are discrete. We have that:*

$$H(\hat{h}) \geq ng(p_n, q_n, l_n), \tag{20}$$

*where $g$ denotes some non-negative function.*

*Proof.* Consider a sequence of pairs $(s_1^0, s_1^1), \ldots, (s_n^0, s_n^1)$, where each $s_i^j = (x_i^j, y_i^j)$ is sampled independently according to the data generating distribution $\mathcal{D}$. Let $E = ((s_i^0, s_i^1))_{i=1,\ldots n}$ denote the of sample pairs. Additionally, let $b_1, \ldots, b_n \in \{0, 1\}$ denote $n$ i.i.d. Bernoulli random variables, and let $B = (b_i)_{i=1,\ldots,n}$ denote the sequence. We may construct a sample $S$ by selecting elements of $E$ according to $B$:

$$S = ((x_i^{b_i}, y_i^{b_i}))_{i=1,\ldots,n}, \tag{21}$$

and we note that $S$ is an i.i.d. sample of size $n$ from the data generating distribution $\mathcal{D}$. Additionally, by independence of $B$ and $E$, we have that $H(B \mid E) = H(B) = n \log 2$. On the other hand, we have

$$H(B \mid E, \hat{h}) \leq \sum_{i=1}^{n} H(B_i \mid E, \hat{h})$$

$$\leq \sum_{i=1}^{n} H(B_i \mid \hat{h}(x_i^0), \hat{h}(x_i^1), y_i^0, y_i^1)$$

$$\leq \sum_{i=1}^{n} H(B_i \mid L(\hat{h}(x_i^0), y_i^0)).$$

We compute the conditional distribution of $B_i$ given $L(\hat{h}(x_i^0), y_i^0) = L_i^0$. In particular, we claim that $B_i \mid L_i^0 = 0$ is Bernoulli with parameter $p_n$. Indeed, note that $B_i$, $L_i^0$ and $\hat{h}$ have the same distribution as if they were sampled from the procedure described before (19). Namely, sample $S$ i.i.d. according to the data generating distribution, and let $\hat{h}$ be the corresponding estimator, $B_i$ an independent Bernoulli random variable, and $L_i = L(\hat{h}(x), y)$ where $(x, y)$ is sampled uniformly from $S$ if $B_i = 0$ and according to the data generating distribution if $B_i = 1$. Note that this distribution does not depend on $i$ due to the assumption that $\hat{h}$ is measurable with respect to the unordered sample $S$. By (19), we thus deduce that:

$$\mathbb{P}(B_i = 0 \mid L_i^0 = 1) = p_n \tag{22}$$

which yields the desired result by taking expectation over the distribution of $\hat{h}, \hat{L}(\hat{h})$.

Similarly, we may compute the distribution of $B_i$ conditional on the event where $L_i^0 = 0$, as $\mathbb{P}(B_i = 0 \mid L_i^0 = 0) = q_n$. By definition, we now have that:

$$H(B_i \mid L_i^0) = l_n h_b(p_n) + (1 - l_n) h_b(q_n), \tag{23}$$

where $h_b(p)$ denotes the binary entropy function. Finally, we apply the chain rule for entropy. We note that

$$H(B \mid E, \hat{h}) = H(B, \hat{h} \mid E) - H(\hat{h} \mid E), \tag{24}$$

and write $H(B, \hat{h} \mid E) \geq H(B \mid E) = n \log 2$ and $H(\hat{h} \mid E) \leq H(\hat{h})$. In summary,

$$
\begin{aligned}
H(\hat{h}) &\geq H(\hat{h} \mid E) \\
&= H(B, \hat{h} \mid E) - H(B \mid E, \hat{h}) \\
&\geq n \log 2 - n[l_n h_b(p_n) - (1 - l_n) h_b(q_n)] \\
&\geq n[h_b(1/2) - l_n h_b(p_n) - (1 - l_n) h_b(q_n)] \,,
\end{aligned}
$$

which yields (20). $\qquad\square$

