# OpenReview forum: "Non-vacuous Generalization Bounds at the ImageNet Scale: a PAC-Bayesian Compression Approach"
_ICLR.cc/2019/Conference_

### Official Review · AnonReviewer2 · 2018-10-22
**Good paper provided the authors have answers to some technical questions.**

**Rating:** 8
**Confidence:** 4

**Review:**

This paper gives the first nonvacuous generalization bounds for
meaningful Imagenet models.  These bounds are given in terms of the
bit length of compressions of learned models together with a method
for taking into account symmetries of the uncompressed parameters.

These bounds are nonvacuous only when the compressed models are small
--- on the order of 500 Kilobytes.  State of the art compressed models
of this size achieve Imagenet accuracies slightly better than Alexnet,
16% error for top 5, and this paper reports a nonvacuous
generalization guarantees of 89% error for top 5.  While there is
still a large gap between the actual generalization and the guarantee,
this would still be a significant accomplishment.

I have one major concern.  The generalization bound involves adding an
empirical loss and a regularization term computed from a KL
divergence.  I am convinced that the authors have correctly handles
the KL divergence term.  But the paper does not contain sufficient
detail to determine if the authors correctly handle the empirical loss
term.  It is NOT correct to use the training loss of the
(deterministic) compressed model.  The generalization bound requires
that the training loss be measured under the parameter noise of the
posterior distribution.  The paper needs to be clear that this has
been done. The comments in Appendix B on noise robustness are
disturbing in this regard.

If the training loss  has been calculated correctly in the bound,
the results are significant.

Assuming correctness, I would comment that the Catoni bound, while sqeaking
out all available tightness, is very opaque.  I might be good to
consider the more transparent bounds, claimed to be essentially the
same, given in McAllester's tutorial.  If the more transparent bounds
achieve equivalent numerical results, they would make the nature of
the bounds clearer.

Another comment involves a largely ignored detail in (Dzuigaite and
Roy 17). Their bounds become vacuous if they center their Gaussian
prior at zero.  Instead they center the prior on the initial value of
the parameters.  This yields a dramatic improvement in the bound.  In
the context of the present paper, this suggests a modification of the
prior distribution on the compressed model.  We represent the model by
first selecting the r code values.  I think a distribution could be
defined on the code book that would improve its log probability, but I
will ignore that.  Given the r code values we can define a
distribution over the possible compressed representations of a weight
w_i in terms of a prior on w_i defined in terms of its initial value.
This gives a probability distribution over the compressed
representation.  Using log probability of the compressed
representation should then be a significant improvement on the first
term in (8).  This shift in the prior on compressed models has no
effect on the second term of (8) so things should only get better.

---

> ### Author Response · Authors · 2018-11-09
> **Thank you for your review**
>
> Thank you for the detailed and insightful review.
>
> As you point out, the empirical loss used in the bound is that of the stochastic classifier. We confirm that we did use the value for the stochastic network, which is indeed slightly worse than that of the non-perturbed network (65% training accuracy vs. 67% training accuracy). We have clarified these details in Appendix B (experimental details).
>
> We agree that Catoni’s bound is unfortunately more opaque than other variants of PAC-Bayes. We have added a remark to the paper noting this and giving references to unified treatments of the different bounds. As we note in the remark, Catoni’s variant is significantly stronger when KL / n is large, as in our case. We provide a comparison of the different bounds here: https://github.com/anonymous-108794/nnet-compression/blob/master/artifacts/plots/README.md In our application for ImageNet, we have that KL / n is approximately 1.5.
>
> Your suggestion about incorporating the initialization weights is very interesting. We have previously experimented with a similar idea, also inspired by Dzugiate and Roy. We represented the weights as the difference between the initial and final values, with the hope that this would afford a more compressed representation. Unfortunately, we were not able to witness clear improvements with such strategies. Your suggestion seems like a promising direction!

---

### Official Review · AnonReviewer3 · 2018-10-30
**a nice bound for the ImageNet**

**Rating:** 6
**Confidence:** 5

**Review:**

The paper presents an application of PAC-Bayesian bounds to the problem of
ImangeNet classification (a deep neural network model). The authors provide
interesting empirical bounds for the risk of the ImageNet classifier. More specifically,
the authors introduce some clever choices for the prior distribution (on the
hypothesis space) that allow one to incoperate a compression scheme and obtain
a (non-vacuous) bound for the predictor.
Overall, This is an original work with clear presentation.

Major comments:
1). In Theorem 2.1, why do you need \lambda > 1 ?
To my knowledge, \lambda only needs to be positive.
Why do you have to introduce the parameter \alpha here?
and consequently the additional log term?
2) It is unclear for me, why are your bounds non-vacuous?
Probably, a more clear explanation of Theorem 4.3 is to be required.
Also, some comparisions with the bounds in [Neyshabur et al 2018] and [Barlett et al 2017]
would make the paper more significant and interesting.

Minor comments:
1) in Theorem 2.1, after the formula (3), the \Phi^{-1} should be  \Phi^{-1}_{\gamma}.
2) in the sentence, page 4,: "To strengthen a naïve Occam bound, we use the idea that that deep networks are insensitive to mild... "   an extra "that" should be removed.
3) in Section 5, the first paragraph, in sentence:  "The lone exception is Dziugaite & Roy (2017), which succeeds by ...."
should be "The one exception...."

---

> ### Author Response · Authors · 2018-11-09
> **Thank you for your review**
>
> Thank you for your positive comments, careful review and insightful questions. We have corrected the typos in the new version of the manuscript. We now address your two questions.
>
> 1) As you correctly state, the bound holds for any fixed lambda (including those smaller than 1). However, the bound is vacuous (its value is larger than 1) when epsilon is small and lambda < 1: indeed, note that phi^{-1}(x) > 1 when x > 1, and the argument is larger than 1 when lambda < 1 and epsilon is small. We introduce alpha (and the log terms) to allow for optimization over lambda. See [Catoni, p. 13], for a full derivation.
>
> 2) We are not certain what the question, “why are your bounds non-vacuous?” means. Could you please elaborate? By non-vacuous, we mean that the obtained generalization error is better than guessing at random (which is 0.999 for top-1 on the 1000 class ImageNet problem). This is not an inherent property of Theorem 4.3, but an observation of the application of the bound in this specific context.
>
> Unfortunately, fair quantitative comparisons with existing bounds are difficult. In particular, many authors do not include constants required to evaluate the bound (e.g. Neyshabur et al. 2018, Theorem 1). To the best of our knowledge, attempts to evaluate these bounds have shown that they are tens of orders of magnitude too large to give non-vacuous bounds in realistic applications (see Arora et al. figure 4),  even ignoring all constants and logarithmic terms.
>
> Neyshabur et al. 2018: https://arxiv.org/abs/1802.05296
> Arora et al. 2018: https://arxiv.org/abs/1802.05296

---

### Official Review · AnonReviewer1 · 2018-11-03
**A very inspiring implementation but too many important details are missing.**

**Rating:** 6
**Confidence:** 4

**Review:**

This paper tries to push forward in important directions the seemingly increasingly powerful approach of using PAC-Bayesian formalism to explain low risks of training neural nets on real-life data. They take an interesting approach to evaluate these bounds by setting up a prior distribution as a mixture of Gaussians centered on possible heuristic compressions of the net  and this prior's variances are obtained by doing a layerwise grid search. This seems to give good risk bounds on certain known compressible nets using image data sets.

Let me list out a bunch of issues that seem to be somewhat confusing in this paper (some of these were in the comment thread I had with the authors but I am repeating nonetheless for completeness)

0.
Firstly this form of the PAC-Bayes formula used here (Theorem 2.1) is of a more complicated form than what has been previously used in say these papers, https://arxiv.org/abs/1707.09564 Given this I strongly feel that there is a need for an explanation connecting this formalism to the usual one - particularly something that proves how this is stronger than the one in the paper I referred to earlier.

1.
In the statement of Theorem 2.1 there is a \lambda parameter over which the infimum is being taken. If I understand right in the experiments one is substituting the upperbound on KL from Theorem 4.3 into this RHS of Theorem 2.1 and evaluating this. Now there is also a \lambda parameter in Theorem 4.3. Is this the same \lambda as in Theorem 2.1 and when a grid-search is being done over \lambda is the "whole" thing (theorem 2.1 upperbound with theorem 4.3 substituted) being minimized by choosing a good \lambda?

If the two \lambda s are different then is the choice of the 2 \lambda s being optimized separately?

(...the authors had earlier clarified that this is so and I strongly feel this is a very important clarification should be updated into the paper..)

2.
How is the \sigma of Theorem 4.3 chosen in the experiments? Am I right in thinking that this \sigma is the posterior variance about which it is being said towards the end of page 6 that "We add Gaussian noise with standard deviation equal to 5% of the difference between the largest and smallest weight in the filter." ?

So am I to understand that this is an arbitrary choice? Or is this choice dictated by some need to ensure that the posterior variance sigma is chosen so that under this distribution the sampled nets approximately compute the same function on the training data? (If yes, then what in the theory is motivating this?).

To the best of my understanding the results are highly dependent on this choice of sigma but there is virtually no explanation for this choice which was not even found by grid search. (As of now this is merely reflective of the fact that trained nets often have some noise resilience but its not a priori clear as to why that should be important to the PAC-Bayes formalism here.)

3.
The code based compression seems a bit mysterious to me given that I do not have enough familiarity with the algorithm that is being referred to. Hence it seems a bit weird as to why there is a sum over codebooks in the proof of Theorems 4.3. Naively I would have thought that there is a fixed codebook for a given compression scheme but here it feels that the compression scheme is a randomized algorithm which also generates a new codebook in every run of it. This seems unusual and seems to need more explanation and at the very least a detailed pseudocode explaining exactly how this compression is working.

This point ties in with a somewhat larger issue I describe next...

4.
In the previous reply to my comment the authors had shared their anonymized code and l had a look through the code. Its pretty evident from the code there are an enormous number of tweaks and hyperparameter tunings to make this work. There is very little insight otherwise as to why "Dynamic Network Surgery' should work and its great that the authors have found an implementation that works on their image data.

But then the question arises that there should have been a cleanly abstracted out pseudocode explaining how the compression was done and how the dynamic network surgery was done. To my mind this implementation is the main contribution of the paper and giving the pseudocode for it in the paper seems not only important for essential completeness of the current paper but that could also then act as a springboard for many future attempts at trying to come up with theory for these mysterious procedures.

---

> ### Author Response · Authors · 2018-11-09
> **Thank you for your review**
>
> Thank you for your careful reading and detailed questions and comments. .
>
> 0. We have added a remark following Theorem 2.1  noting that this form is relatively complicated, explaining the reason we use it, and providing references to a unified treatment of the different PAC-Bayes bounds. In particular, Laviolette (slide 16) gives a general formulation which encompasses all existing formulas. Catoni’s formulation is significantly tighter for large values of KL, which is the case in our paper. We provide a plot comparing the different bounds here: https://github.com/anonymous-108794/nnet-compression/blob/master/artifacts/plots/README.md In our application to ImageNet, we have that KL / n is approximately 1.5.
>
>
> 1. Thank you for pointing out the notational overload. In the revised version, we have adjusted the notation in Theorem 4.3 (the prior variance is now called tau) to prevent confusion, as the two lambdas were indeed distinct. We have also added a section in the appendix (A.1) to better explain how the bound is adjusted for this choice.
>
> 2. The posterior variance sigma is chosen as to provide significant improvement in the bounds (by witnessing noise robustness) while minimally affecting the performance of the estimator. Sigma is part of the posterior and can be chosen in an arbitrary (data dependent) manner without affecting the validity of the bound. We choose sigma to minimally affect the estimator to ensure that bounds on the stochastic estimator are reflective of the performance of the deterministic estimator. We have included some more details to this effect in Appendix B.
>
> 3 and 4:
> We agree that a main contribution of the paper is the implementation that allows us to demonstrate that the compression-generalization link has real explanatory power for realistic deep learning applications. A strength of our bound is that it is compatible with any compression scheme. The particular the compression strategy we use was chosen because it was state of the art for compression at time of writing. We use the strategy of Han et al. (2016) with the pruning method of Guo et al. (2016). We do not make modifications to the procedures they describe, beyond hyperparameter tuning.
>
> We anticipate that better neural network compression schemes will be developed, and future work can use our work with these better compression schemes. Accordingly, we feel that reproducing detailed descriptions of the particular compression scheme would be somewhat misleading. The respective authors provide pseudocode and detailed explanations in their original papers.
>
> As an aside, we started with the deep compression scheme of Han et al. (2016)  and modified the pruning strategy to that of Guo et al. (2016). The pruning schedule was taken from Zhu and Gupta (2018) and final sparsity values were inspired by Iandola et al. (2016).
>
> Laviolette: https://bguedj.github.io/nips2017/pdf/laviolette_nips2017.pdf

---

### Comment · AnonReviewer1 · 2018-10-23
**Request for some clarifications**

I request a few clarifications from the authors to help review this paper.

1.
In the statement of Theorem 2.1 there is a \lambda parameter over which the infimum is being taken. If I understand right, in the experiments one is substituting the upperbound on KL from Theorem 4.3 into this RHS of Theorem 2.1 and is evaluating this. Now there is also a \lambda parameter in Theorem 4.3. Is this the same \lambda as in Theorem 2.1 and when a grid-search is being done over \lambda is the "whole" thing (theorem 2.1 upperbound with theorem 4.3 substituted) being minimized by choosing a good \lambda?

(At some point the paper says that "we choose the prior variance \lambda^2 layerwise by a grid search". Does this mean that the formula being computed in the code uses a different \lambda for each layer and hence its not Theoem 4.3's RHS that is being computed in the code?)

If the two \lambda s are different then is the choice of the 2 \lambda s being optimized separately?

2.
How is the \sigma of Theorem 4.3 chosen in the experiments? Am I right in thinking that this \sigma is the posterior variance about which it is being said towards the end of page 6 that "We add Gaussian noise with standard deviation equal to 5% of the difference between the largest and smallest weight in the filter." ?

So am I to understand that this is an arbitrary choice of \sigma or something has been optimized to get this? Or was this choice of \sigma constrained by wanting that the stochastic net have w.h.p almost the same function values as the original trained net? If yes, then from where in the theory is such a constraint arising from?

3.
The footnote of page 6 says, "Code to reproduce the experiments is available in the supplementary material." but I dont see any such thing anywhere. Neither is any pseudocode available to check.

4.
At the very end of section 5 one finds this line, "stochastic network has a top-1 accuracy of 65 % on the training data, and top-5 accuracy of 87 % on the training data". So this means that your top-1 training error is 35% and top-5 training error is 13%. And I guess the second row of your table is what corresponds to this where you claim that your numerically optimized theoretical bound is giving upperbounds of 96.5% and 87% respectively. Am I right?

Then the question arises as to where and how in the theory (the combination of Theorem 2.1 and 4.3?) were you able to specify that it should evaluate the top-1 and top-5 error?  The paper does not seem to specify any loss function either where such a thing can be incorporated.

---

> ### Author Response · Authors · 2018-10-23
> **Clarifications for Reviewer 1**
>
> Thank you for the detailed reading of the paper and the comprehensive questions.
>
> 1. Your description of the selection procedure for lambda is correct, along with your description of the procedure
> for several lambdas (they are optimized separately, which is equivalent to optimizing jointly as the upper bound is separable). You are correct that due to the selection, we are not directly applying Theorem 4.3, but also combining it with a union bound to ensure correctness. One way to view it is the following: let \pi_\lambda denote the prior distribution in Theorem 4.3 with \lambda fixed. We can define a new prior \pi, which is the uniform mixture of \pi_\lambda for \lambda varying over all 2^32 values corresponding to IEEE-754 single precision floating point numbers. Let \pi_\lambda* denote the prior selected by our grid search. Then we have that: \pi_\lambda*(x) \leq \pi_\lambda / 2^32, and hence KL(\rho, \pi) \leq KL(\rho, \pi_\lambda^*) + 32 \log 2. We apply the PAC-Bayesian bound with the prior \pi instead of \pi_\sigma, and use the above bound (note: in practice we select a lambda for each layer, thus selecting 20 or so parameters, a similar argument apply). The cost paid in terms of KL divergence is thus 32 bit for each parameter, or less than 1000 bits in total, which is negligible (but taken into account) compared to the total effective size - we have thus not included this detail, although we can certainly clarify in the appendix if necessary.
>
> 2. The value of \sigma is chosen "by wanting that the stochastic net have w.h.p. the same function values [performance] as the original net". There is no constraint from a theoretical perspective in the choice of \sigma (as it is part of the posterior, it can be chosen in an arbitrary, including data-dependent, manner). Our choice of sigma captures the intuition that neural networks tend to be somewhat robust to low levels of noise.
>
> 3. Unfortunately, due to technological constraints, we were unable to upload the supplementary material to ICLR. We have created an anonymized Github repository with the code at: https://github.com/anonymous-108794/nnet-compression.
>
> 4. Your interpretation of table 1 is right. The theory (Theorem 2.1) can be applied to any {0,1}-valued loss function, which includes both top-1 and top-5 accuracy (which is equal to 1 if the true label is in the top-1 (resp. top-5) most likely predicted labels, and 0 otherwise). We have chosen these two metrics as they are the most commonly used metrics on ImageNet.

---

### Meta-Review · Area_Chair1 · 2018-12-18
**Nice paper with novel results**

**Confidence:** 4
**Recommendation:** Accept (Poster)

**Metareview:**

The paper combines PAC-Bayes bound with network compression to derive a generalization bound for large-scale neural nets such as ImageNet. The approach is novel and interesting and  the paper is well-written. The authors provided detailed replies and improvements in response to reviewers questions, and all reviewers agree this is a very nice contribution.